# Honeybee Venom Synergistically Enhances the Cytotoxic Effect of CNS Drugs in HT-29 Colon and MCF-7 Breast Cancer Cell Lines

**DOI:** 10.3390/pharmaceutics14030511

**Published:** 2022-02-25

**Authors:** Diana Duarte, Soraia I. Falcão, Iouraouine El Mehdi, Miguel Vilas-Boas, Nuno Vale

**Affiliations:** 1OncoPharma Research Group, Center for Health Technology and Services Research (CINTESIS), Rua Doutor Plácido da Costa, 4200-450 Porto, Portugal; dianaduarte29@gmail.com; 2Faculty of Pharmacy, University of Porto, Rua Jorge Viterbo Ferreira, 228, 4050-313 Porto, Portugal; 3Centro de Investigação de Montanha (CIMO), Instituto Politécnico de Bragança, Campus de Santa Apolónia, 5300-253 Bragança, Portugal; sfalcao@ipb.pt (S.I.F.); iouraouine@gmail.com (I.E.M.); mvboas@ipb.pt (M.V.-B.); 4Department of Community Medicine, Health Information and Decision (MEDCIDS), Faculty of Medicine, University of Porto, Alameda Professor Hernâni Monteiro, 4200-319 Porto, Portugal

**Keywords:** honeybee venom, melittin, colorectal cancer, breast cancer, drug combination, antineoplastic drugs, CNS drugs, drug repurposing

## Abstract

5-fluorouracil (5-FU) and doxorubicin (DOX) are potent anti-tumour agents commonly used for colon and breast cancer therapy, respectively. However, their clinical application is limited by their side effects and the development of drug resistance. Honeybee venom is a complex mixture of substances that has been reported to be effective against different cancer cells. Its active compound is melittin, a positively charged amphipathic peptide that interacts with the phospholipids of the cell membrane, forming pores that enable the internalization of small molecules with cytotoxic activities,^.^ and consequently, causing cell death. Some central nervous system (CNS) drugs have recently demonstrated great anti-cancer potential, both in vitro, in vivo and in clinical trials, being promising candidates for drug repurposing in oncology. The present work evaluated the anti-cancer efficacy of honeybee venom in combination with chemotherapeutic or CNS drugs in HT-29 colon and MCF-7 breast cancer cell lines. The chemical characterization of a Portuguese sample of honeybee venom was done by LC-DAD-ESI/MSn analysis. For single treatments, cells were incubated with increasing concentrations of bee venom. For combination treatments, increasing concentrations of bee venom were first combined with the half-maximal inhibitory concentration (IC_50_) of 5-FU and DOX, in HT-29 and MCF-7 cells, respectively. Cells were also treated with increasing concentrations of bee venom in combination with the IC_50_ value of four CNS drugs (fluphenazine, fluoxetine, sertraline and thioridazine). Cytotoxicity was evaluated by MTT and SRB assays. The combination index (CI) value was calculated using CompuSyn software, based on the Chou–Talalay method. Synergy scores of different reference models (HSA, Loewe, ZIP and Bliss) were also calculated using SynergyFinder. The results demonstrate that honeybee venom is active against HT-29 colon and MCF-7 breast cancer cells, having better anti-tumour activity in MCF-7 cells. It was found that bee venom combined with 5-FU and fluphenazine in HT-29 cells resulted in less cytotoxic effects compared to the co-treatment of fluoxetine, sertraline and thioridazine plus bee venom, which resulted in less than 15% of viable cells for the whole range of concentrations. The combination of MCF-7 cells with repurposed drugs plus honeybee venom resulted in better anti-cancer efficacies than with DOX, notably for lower concentrations. A combination of fluoxetine and thioridazine plus honeybee venom resulted in less than 40% of viable cells for all ranges of concentrations. These results support that the combination of honeybee venom with repurposed drugs and chemotherapeutic agents can help improve their anti-cancer activity, especially for lower concentrations, in both cell lines. Overall, the present study corroborates the enormous bioactive potential of honeybee venom for colon and breast cancer treatments, both alone and in combination with chemotherapy or repurposed drugs.

## 1. Introduction

Colorectal cancer (CRC) and breast cancer are among the most prevalent types of cancer worldwide. CRC represents the third most common cause of cancer-related deaths in the United States of America (USA) while breast cancer has the highest incidence among all cancers, being the second most common cause of death in women around the world [1]. Chemotherapy plays an important role in cancer therapy. 5-fluorouracil (5-FU) is the most commonly used antineoplastic drug for the treatment of CRC, after surgical removal of the tumour. Nevertheless, due to its short half-life, high cytotoxicity, and low bioavailability, 5-FU use is very limited [2]. Doxorubicin (DOX) is an anthracycline drug and is widely used as a chemotherapeutic agent for breast cancer therapy. Despite its great anti-cancer efficacy, its use is limited by cardiotoxicity [3]. Due to the lack of selectivity of antineoplastic agents, higher doses of chemotherapeutics are usually necessary to effectively kill cancer cells, resulting in toxicity to normal cells and severe side effects to the patients, along with the development of drug resistance [4]. Thus, the development of new strategies to decrease chemotherapeutic drugs’ toxicity and overcome drug resistance are urgently desired.

The discovery of new drugs for combatting cancer is a complicated process in terms of time and costs, which usually results in a low probability of the new drug to enter Phase I clinical trials [5]. Drug combination is a strategy that can help improve the efficacy of cancer treatments and consists of combining two or more therapeutic agents. This strategy takes advantage of the overlapping mechanisms of action of each drug to achieve synergism, i.e., a potentiation of the therapeutic effect, allowing a decrease in the therapeutical dose, and consequently, the side effects of the drugs [6]. Several studies have already demonstrated that drug combination is more effective than monotherapy [7,8,9,10]. Another strategy that can generate novel candidates for combatting cancer is drug repurposing. This strategy uses already FDA-approved drugs and evaluates them for new indications other than the original, saving time and money while ensuring the necessary safety and toxicity profiles [11,12]. A review from 2019 demonstrated that some central nervous system (CNS) drugs show great potential as anti-cancer drugs both in vitro, in vivo and clinical trials [13]. Our most recent study also found some CNS drugs with promising anti-cancer profiles against colon and breast cancer cells [14]. The most promising ones were fluphenazine, fluoxetine, sertraline and thioridazine. Fluphenazine and thioridazine act by blocking postsynaptic mesolimbic dopaminergic D1 and D2 receptors in the brain and depressing the release of hypothalamic and hypophyseal hormones [15,16]. Sertraline and fluoxetine are selective inhibitors of serotonin reuptake at the presynaptic neuronal membrane, thereby increasing serotonergic activity and enhancing the levels of 5-hydroxytryptamine (5-HT) in the brain [17,18].

Honeybees secrete bee venom (BV) from a highly specific venom gland connected to the venom reservoir located in the abdominal cavity. Mainly composed of water, with a composition of 80%, bee venom is a complex mixture containing at least 18 pharmacological active compounds, which includes peptides like melittin, apamin, mast cell degranulating (MCD) peptides, adolapin, enzymes such as phospholipase A2 (PLA2) and hyaluronidase, bioactive amines such as histamine and dopamine, amino acids, minerals and volatile compounds [19]. With a long tradition of applications in natural medicines to treat diseases such as arthritis, rheumatism, pain, tumours, and skin diseases, several studies describe the therapeutic potential of bee venom components as anti-inflammatory, anti-viral and anti-cancer [19,20]. Recent research correlated bee venom to a variety of cancer management effects including induction of apoptosis, necrosis, cytotoxicity and inhibition of proliferation in different cancer types, including prostate, breast, lung, liver and bladder [20]. The rich composition in venom peptides, which are described to exhibit high specificity and selectivity towards cancer cells, with effects on cell proliferation, invasion, migration, and angiogenesis, as well as modulating immune responses [21], shows the increased potential of bee venom for targeted cancer therapy.

In this work, we chemically characterized a honeybee venom sample and investigated the differential modulations of the treatments either with chemotherapeutics or CNS drugs alone, or combined with honeybee venom on cellular viabilities in the HT-29 human colon and MCF-7 breast cancer cells. We designed a combination model consisting of increasing concentrations of honeybee venom plus a fixed dose of a chemotherapeutic agent. 5-FU and DOX were used as reference drugs for HT-29 and MCF-7 cancer cells, respectively. Increasing concentrations of honeybee venom were also combined with a fixed concentration of four different CNS drugs (fluphenazine, fluoxetine, sertraline and thioridazine) to assess if repurposed drugs have an increased potential for drug combination than the chemotherapeutic drugs for these cell lines. These combinations were also assessed for drug synergism, to evaluate if there was a potentiation of the effect compared to both drugs alone. The aim of combining honeybee venom with these drugs was to enhance the anti-tumour effect of chemotherapeutic and repurposed drugs for colon and breast cancer therapy, using a combination model consisting of a natural compound plus a therapeutic drug, instead of using two therapeutic drugs, as is usually done in drug combination studies for combatting cancer.

Our results demonstrate that honeybee venom increased the efficiency of chemotherapeutic and CNS drugs in MCF-7 breast and HT-29 colon cancer cell lines, with an indication of synergic interactions, especially for lower concentrations. These drug combinations have a great potential to be further considered as adjuvant therapies for colon and breast cancer.

## 2. Materials and Methods

### 2.1. Chemical Characterization of the Samples

#### 2.1.1. Reagents and Materials

Apamin from bee venom (purity 98.3%) was obtained from CalBiochem (San Diego, CA, USA). Phospholipase A2 from bee venom (activity: 1775 units mg^−1^ solid), cytochrome c from equine heart (purity ≥ 95%) and melittin from bee venom (purity: ≥85%) were obtained from Sigma-Aldrich (St. Louis, MO, USA). Formic acid and HPLC grade acetonitrile were obtained from Fisher Scientific UK (Loughborough, Leics, UK). Water was treated in a Milli-Q water purification system (TGI Pure Water Systems, Brea, CA, USA).

#### 2.1.2. Honeybee Venom

The bee venom was gathered in 2018 from *Apis mellifera iberiensis* hives located in Bragança, Portugal. The sample was collected using a double-face bee venom collector, developed in our facilities with some specific features. The device was placed in the hives at one of the outmost opposite ends of the beehive nest. A mild electrical impulse shock was applied (increasing from 0 V to 12 V then decreasing to initial voltage). The optimum duration of bee venom collection in the beehive was set between 30 min to 60 min, early in the morning or at the beginning of the sunset. After the collection period, the venom was scraped off from the glass with a sharp scraper and conditioned in pharmaceutical-grade vials. The bee venom was then freeze-dried at −20 °C until further analysis.

#### 2.1.3. LC-DAD-ESI/MS^n^ Analysis

The LC-DAD-ESI/MS^n^ study was performed on a Dionex Ultimate 3000 UPLC instrument (Thermo Scientific, San Jose, CA, USA) equipped with a diode-array detector and coupled to a mass detector. The chromatographic system consisted of a quaternary pump, an autosampler maintained at 5 °C, a degasser, a photodiode-array detector, and an automatic thermostatic column compartment. Chromatographic separation was carried out according to previously described work [22] on an XSelect CSH130 C18, 100 mm × 2.1 mm id, 2.5 µm XP column (Waters, Milford, MA, USA), and its temperature was maintained at 30 °C. Spectral data of all peaks were accumulated in the range of 190–500 nm. Cytochrome *c* was used as internal standard (IS) and prepared in deionized water at the concentration of 25 µg/mL. For the analysis, the lyophilized HBV (3 g) was dissolved in 10 mL of IS solution. Each sample was filtered through a 0.2 µm nylon membrane (Whatman). The standard solutions were prepared by dissolving the compound in IS solution, at the desired concentration. Quantification was achieved using calibration curves for apamin (4–60 µL/mL; *y* = 0.031*x* + 0.021; *R*^2^ = 0.998), PLA2 (8–120 µL/mL; *y* = 0.026*x* + 0.118; *R*^2^ = 0.996) and melittin (31–500 µL/mL; *y* = 0.034*x* + 0.026; *R*^2^ = 0.999). The mass spectrometer was operated in the positive ion mode using a Linear Ion Trap LTQ XL mass spectrometer (Thermo Scientific, San Jose, CA, USA) equipped with an ESI source. Typical ESI conditions were nitrogen sheath gas 35 psi, spray voltage 3.5 kV, source temperature 300 °C, capillary voltage 20 V, and the tube lens offset was kept at a voltage of 74 V. The collision energy used was 30 (arbitrary units). Data acquisition was carried out with the Xcalibur^®^ data system (Thermo Scientific, San Jose, CA, USA).

### 2.2. Cytotoxic Studies

#### 2.2.1. Reagents and Materials

McCoy’s 5A Medium powder (modified with L-glutamine, without sodium bicarbonate), Dulbecco’s modified Eagle’s medium (DMEM), fetal bovine serum (FBS) and penicillin-streptomycin (pen-strep) solution were purchased from Millipore Sigma (Merck KGaA, Darmstadt, Germany). Other cell culture reagents were purchased from Gibco (Thermo Fisher Scientific, Inc, Waltham, MA, USA). Thiazolyl blue tetrazolium bromide (MTT, cat. no. M5655), sulforhodamine B (SRB, cat. no. S1402), 5-fluorouracil (5-FU, cat. no. F6627) and fluphenazine dihydrochloride (cat. no. F4765) were obtained from Sigma-Aldrich (Merck KGaA, Darmstadt, Germany). Doxorubicin hydrochloride (DOX, cat. no. 15007), thioridazine hydrochloride (cat. no. 14400), sertraline hydrochloride (cat. no. 14839) and fluoxetine hydrochloride (cat. no. 14418) were obtained from Cayman Chemical (Ann Arbor, MI, USA).

#### 2.2.2. Cell Culture

Two different cell lines were used in this study (MCF-7 breast and HT-29 colon cancer cell lines). Both cell lines were obtained from the American Type Culture Collection (ATCC; Manassas, VA, USA) and maintained, according to ATCC’s recommendations, at 37 °C and 5% CO_2_. HT-29 cells were grown in McCoy’s 5a Medium Modified and MCF-7 cells were maintained in DMEM cell culture medium. All media were supplemented with 10% FBS and a 1% pen-strep solution. Cells were passaged using a 0.25% trypsin-EDTA solution when confluency reached 80–90% and were subcultured in the same culture media. Prior treatments, HT-29 cells (15,000 cells/well) and MCF-7 cells (10,000 cells/well) were seeded in 96-well plates and allowed to adhere overnight.

#### 2.2.3. Drug Treatment

The cytotoxic effect of 5-FU, DOX, four different CNS drugs (fluphenazine, fluoxetine, sertraline and thioridazine) and bee venom for single and combination studies was evaluated after 48 h of treatment. First, the half-maximal inhibitory concentration (IC_50_) value was determined for each drug alone both in HT-29 and MCF-7 cells. 5-FU, DOX and CNS drugs’ concentrations ranged from 0.1 to 100 µM for single-drug treatments. Bee venom was tested alone in concentrations of 6.25, 12.5, 25, 50 and 100 µg/mL. Additionally, combination studies were performed by combining 5-FU, DOX or each CNS drug (Drug 1) at their IC_50_ value with bee venom at the same range of concentrations tested for the single treatment (Drug 2). Control cells were treated with 0.1% DMSO (vehicle). The combined effects of non-equipotent concentrations (non-fixed ratio) were evaluated by MTT and SRB assays.

#### 2.2.4. Cell Viability Assays (MTT and SRB)

The effects of single treatment with 5-FU, DOX, CNS drugs and bee venom on the viability of MCF-7 and HT-29 cells were evaluated by an MTT assay. The effects of combined treatments were evaluated by MTT and SRB assays. For the MTT protocol, after each treatment, cell media were aspirated and 100 µL of MTT solution (0.5 mg/mL in PBS) was added to each well. Cells were protected from light and incubated for a period of 3 h. Then, MTT solution was aspirated from each well and replaced with 100 µL/well of DMSO to solubilize the formazan crystals. Absorbance was measured at 570 nm using an automated microplate reader (Tecan Infinite M200, Tecan Group Ltd., Männedorf, Switzerland). The IC_50_ was determined for each drug and was defined as the concentration showing 50% of cell growth inhibition, compared to control. SRB assay was performed in parallel with MTT assay and was initiated by fixing the cultured cells with ice-cold 10% trichloroacetic acid (TCA) for 30 min after each treatment. Then, cells were stained with 0.4% SRB for 1 h at room temperature. Excess dye was removed by washing the 96-well plates several times with tap water. The protein-bound dye was dissolved using 200 µL of 10 mM Tris base solution. The determination of absorbance was done using a microplate reader with a filter wavelength of 540 nm (Tecan Infinite M200, Tecan Group Ltd., Männedorf, Switzerland). All conditions were performed three times independently, in triplicate.

#### 2.2.5. Evaluation of Cell Morphology

After each treatment, cell morphology was evaluated using a Leica DMI 6000B microscope equipped with a Leica DFC350 FX camera (Leica Microsystems, Wetzlar, Germany). Images were then analysed with the Leica LAS X imaging software (v3.7.4) (Leica Microsystems, Wetzlar, Germany).

#### 2.2.6. Analysis of Drug Interactions

To quantify drug interactions between 5-FU/DOX/CNS drugs and bee venom, two softwares were used: CompuSyn and SynergyFinder. All simulations were performed assuming that the two drugs were combined in a non-fixed ratio of doses with a fixed concentration of Drug 1 and variable concentrations of Drug 2. CompuSyn (ComboSyn, Inc., New York, NY, USA) was used to estimate the combination index (CI), a quantitative representation of pharmacological interactions. This reference model is based on the unified theory introduced by Chou and Talalay [23]. CI was plotted on the *y*-axis as a function of effect level (Fa) on the *x*-axis to assess drug synergism between drug combinations. A CI < 1 indicated synergism, a CI = 1 indicated an additive interaction and a CI > 1 indicated antagonism. The SynergyFinder software was used to estimate the synergy scores of the proposed drug pairs by four different reference models: Bliss, Loewe, highest single agent (HSA) and zero interaction potency (ZIP). Synergy scores <−10 meant that the interaction between two drugs was likely to be antagonistic; scores from −10 to 10 meant the interaction between two drugs was likely to be additive; and scores >10 meant the interaction between two drugs was likely to be synergistic.

#### 2.2.7. Data Analysis

All experiments were conducted in triplicate (*n* = 3) with 3 replicates. GraphPad Prism 8 (GraphPad Software Inc., San Diego, CA, USA) was used to generate concentration–response curves. This was performed by non-linear regression analysis and the viability of cells treated with each drug was normalized to the viability of control cells. Then, the cell viability fractions were plotted against drug concentrations in the logarithmic scale.

#### 2.2.8. Statistical Analysis

The results are presented as the mean ± SEM for n experiments performed. All data were assayed in three independent experiences, in triplicate. Statistical comparisons between control and treatment groups, at the same time point, were performed with Student’s *t*-tests and one-way ANOVA tests. Statistical significance was accepted at *p* values < 0.05.

## 3. Results

### 3.1. Chemical Characterization of the Honeybee Venom

The chemical characterization of the honeybee venom from *Apis mellifera iberiensis* is shown in Figure 1. The peptide melittin was the most abundant compound with a concentration of 65.6 ± 0.3%, followed by the enzyme phospholipase A2, with 4.0 ± 0.2% and the peptide apamin with 1.2 ± 0.1%.

Melittin has been reported by different authors as the main constituent of BV, accounting for 40–50% of its total dry weight [20,22].

### 3.2. Cell-Based Assays Using HT-29 Cells

#### 3.2.1. Anti-Cancer Activity of Honeybee Venom in HT-29 Colon Cancer Cells

To examine the anti-cancer effect of honeybee venom, HT-29 colon cancer cells were treated for a period of 48 h with increasing concentrations of bee venom, and cell viability was evaluated by MTT and SRB assays. These cell-based assays are among the most commonly used in pharmacological studies involving cancer cell lines to evaluate the anti-cancer effect of different chemotherapeutic agents and provide important information regarding mitochondrial metabolism and cellular protein content. MTT assays measure the metabolic activity of cells, while SRB assays use an acidic protein-dye (SRB) that binds electrostatically and pH-dependent to the basic amino acid residues of the proteins presented in the fixed cells, giving information about the cellular protein content. These assays were also included in our previous combination study [14] and we have kept the same methodological design. Control cells were treated with 0.1% DMSO, the vehicle used for stock preparation. MTT results demonstrated that cells treated with honeybee venom in concentrations above 25 µg/mL had a significant reduction of viability (Figure 2A). The SRB assay demonstrated that cellular protein content was significantly reduced in cells treated with concentrations of bee venom above 6.25 µg/mL (Figure 2B). Based on MTT results, a concentration–response curve was plotted and an IC_50_ value of 24.25 µg/mL was obtained for bee venom (Figure 2C). Treatment with concentrations above 50 µg/mL of bee venom resulted in more than 80% cell reduction in both cell-based assays, demonstrating the great potential of this natural substance to reduce the viability of HT-29 colorectal cancer cells.

#### 3.2.2. Cytotoxic Activity of 5-FU and CNS Drugs in HT-29 Cancer Cells

We next treated HT-29 cells with 5-FU and four different CNS drugs (fluphenazine, fluoxetine, sertraline and thioridazine) in order to find their IC_50_ value to use in the combination models. Cells were treated with increasing concentrations (0–100 µM) of each drug for 48 h and their cytotoxic effect was evaluated by MTT assay. 5-FU is an antineoplastic drug commonly used in the chemotherapy of CRC. CNS drugs were included in our combination models based on our recent findings, where it was demonstrated that these drugs are able to reduce the viability of colon cancer cells in vitro [14].

All tested CNS drugs demonstrated the ability to decrease the viability of HT-29 cells, with a greater anti-cancer effect than 5-FU (Figure 3A). After plotting these results using the GraphPad software, we obtained dose–response curves and calculated the relative IC_50_ values (Figure 3B). These values represent the concentration that reduces the cell viability by 50% between the top and the bottom plateaus of the obtained dose–response curve, after the normalization of the data and non-linear regression analysis. Fluphenazine demonstrated the highest anti-tumour potency with an IC_50_ of 1.86 µM, while fluoxetine IC_50_ was around 6 µM, the highest value among all CNS drugs. Taken together, all CNS drugs tested in HT-29 colon cancer cells resulted in an IC_50_ under 10 µM, which reinforced that CNS drugs are promising drugs for repurposing in the chemotherapy of CRC, and are good candidates for employment in combination therapies.

#### 3.2.3. In Vitro Anti-Cancer Activity of Honeybee Venom Combined with 5-FU and CNS Drugs

After demonstrating the potential of CNS drugs for drug repurposing, we decided to evaluate if honeybee venom could help to further improve the anti-cancer activity of these CNS agents (fluphenazine, fluoxetine, sertraline and thioridazine). To do so, we designed a combination model with variable doses (6.25–100 µg/mL) of honeybee venom combined with a fixed dose (IC_50_ value) of each drug. A combination of honeybee venom with 5-FU was also performed for comparison of anti-cancer efficacy. Cells were treated with each combination for 48 h and cell viability was evaluated by MTT and SRB assays. Morphology analysis was also performed in parallel with these cell-based assays.

MTT results demonstrated that honeybee venom in concentrations above 25 µg/mL had greater anti-cancer potential than 5-FU (at the concentration of its IC_50_). Regarding the combination of honeybee venom with 5-FU, these results demonstrated an improved cytotoxic effect compared to single treatments with bee venom for concentrations ranging from 6.25 to 25 µg/mL. Particularly, for the combined pairs of 5-FU + 12.5 µg/mL of bee venom and 5-FU + 25 µg/mL of bee venom, results were even better than both 5-FU and bee venom alone (Figure 4B). The results of the SRB assay demonstrated that cellular protein content was significantly reduced when 5-FU was combined with 6.25 and 12.5 µg/mL of honeybee venom, compared to 5-FU and bee venom alone for the same concentrations (Figure 4C). Morphological analysis also confirmed that the combination of 5-FU + bee venom was promising in decreasing the number of tumour cells, especially when comparing lower concentrations (Figure 4A).

Morphological analysis of HT-29 cells treated with fluphenazine plus increasing concentrations of honeybee venom resulted in a higher decrease in viable cells compared to bee venom alone, for the same range of concentrations (Figure 5A). Combinations with fluphenazine resulted in enhanced cytotoxic effects, with fluphenazine + 25 µg/mL of bee venom causing higher cytotoxicity than both corresponding drugs alone (Figure 5B). Cellular protein content was significantly reduced in HT-29 cells treated with fluphenazine +6.25 µg/mL and fluphenazine + 12.5 µg/mL, compared to both compounds alone (Figure 5C).

The combination of fluoxetine and honeybee venom caused a higher reduction in the number of HT-29 viable cells (Figure 6A), compared to the single treatment with bee venom. Cell viability was significantly decreased in cells treated with a mixture of 6.25, 12.5 and 25 µg/mL of bee venom plus fluoxetine, being even lower than either drug alone (Figure 6B). Protein content inside cells in cells treated with the combination of 6.25 and 12.5 µg/mL of bee venom + fluoxetine was also lower than cells treated with bee venom or fluoxetine alone (Figure 6C).

Sertraline combined with honeybee venom also resulted in a significant decrease in cell viability, both seen by morphological analysis (Figure 7A) and MTT assay (Figure 7B). MTT results demonstrated that 6.25, 12.5 and 25 µg/mL of honeybee venom combined with sertraline produced better anti-cancer efficacy than each drug alone. For higher concentrations, results were significantly improved compared to sertraline alone. Cellular protein content in combined treatments was significantly lower than sertraline alone for all range of concentrations and lower than both drugs alone for the concentrations of 6.25 and 12.5 µg/mL of honeybee venom plus sertraline (Figure 7C).

The last repurposed drug used in HT-29 combination studies was thioridazine. These combinations produced similar results to sertraline and fluoxetine plus honeybee venom, both by MTT and SRB assays. Compared to bee venom alone, morphological analysis revealed a higher reduction of viable cells when thioridazine was combined with bee venom. These differences were seen mostly for lower concentrations of bee venom (Figure 8A). These results are in agreement with MTT assay results, where it was found that the combination of thioridazine and honeybee venom in concentrations of 6.25, 12.5 and 25 µg/mL caused a significant reduction in viable cells, compared to each drug alone. For higher concentrations, the anti-cancer activity of the combinations was similar to bee venom alone (Figure 8B). Evaluating the protein content of the cells treated with combinations of honeybee venom plus thioridazine also revealed a significant decrease for the two lowest concentrations (6.25 and 12.5 µg/mL of bee venom + thioridazine), compared to thioridazine and bee venom alone (Figure 8C).

Together, these results demonstrate that honeybee venom has anti-cancer activity against colon cancer cells, causing a significant reduction in cell viability in concentrations above 25 µg/mL. CNS drugs (fluphenazine, fluoxetine, sertraline and thioridazine) also demonstrated great potential in reducing the number of viable cells than with 5-FU, a common antineoplastic drug used for CRC. The combination of 6.25, 12.5 and 25 µg/mL of honeybee venom with the repurposed drugs fluoxetine, sertraline and thioridazine resulted in a significant reduction in the viability of HT-29 colon cancer cells compared to each drug alone, demonstrating that bee venom can help improve the anti-cancer activity of CNS drugs in colon cancer cells.

#### 3.2.4. Evaluation of Drug Interaction in HT-29 Cells Treated with Honeybee Venom Combined with 5-FU and CNS Drugs

To evaluate the nature of the drug interactions in the previous combinations, the CompuSyn software was used, based on the Chou–Talalay method. This software calculated the combination index (CI) of each combination and it correlates with synergism (CI < 1), additivity (CI = 1) or antagonism (CI > 1). The CI values of 5-FU and honeybee venom in combination were mostly lower than one, except for one pair, suggesting that the growth inhibitory effect of these compounds in combination was mostly synergistic in HT-29 cells. The same behaviour was seen for combinations involving fluoxetine, sertraline and thioridazine (Figure 9). Regarding the combination of fluphenazine and honeybee venom, it was found only two pairs presented synergism, while most of them were antagonist (Figure 9). Results obtained in the CompuSyn software are summarized in Table 1.

Drug interactions in each combination were also analysed by other four reference models: ZIP, Loewe, Bliss and HAS using the SynergyFinder software [6]. The ZIP model evaluates the drug interactions’ relationships by comparing the change in the potency (effect at certain dose level) of the dose–response curves between individual drugs and their combinations, assuming that two non-interacting drugs were expected to incur minimal changes in their dose–response curves [24]. The Loewe additivity model calculated the expected response as if both drugs were the same [25]. The Bliss independence model assumed a stochastic process in which two drugs elicit their effects independently, and the expected combination effect can be calculated based on the probability of independent events [25]. The HSA model is one of the simplest reference models, which states that the expected combination effect is the maximum of the single drug responses at corresponding concentrations [26]. Synergy scores under −10, from −10 to 10 and above 10 indicated antagonism, additivity and synergism, respectively. Synergy plots regarding the combination of 5-FU (Appendix A), fluphenazine (Appendix A) and fluoxetine (Appendix A) plus honeybee venom indicate mostly additivity, by all reference models. Analysing the synergy plots of sertraline plus honeybee venom (Appendix A), it was found that the drug interaction in this combination was synergistic by ZIP, Bliss and HAS models, with synergy scores of 12.04, 13.52 and 17.25, respectively. Thioridazine combined with honeybee venom was found to be predominantly additive, for all reference models (Appendix A).

### 3.3. Cell-Based Assays Using MCF-7 Cells

#### 3.3.1. Cytotoxic Activity of Honeybee Venom in MCF-7 Breast Cancer Cells

Given the promising results obtained using HT-29 colon cancer cells, we next repeated the same experiments in another cancer cell line, the MCF-7 breast cancer cell line, to evaluate if these drug combinations would also be promising to apply in further research in breast cancer therapy, as it is also one of the most prevalent types of cancer worldwide. We analysed the anti-cancer effect of honeybee venom in MCF-7 for 48 h and found that this compound was able to reduce the cell viability of these cancer cells significantly in concentrations above 25 µg/mL (Figure 10A). This compound was also able to reduce the cellular protein content of MCF-7 breast cancer cells in concentrations above 12.5 µg/mL (Figure 10B). Based on MTT results, a dose–response curve was plotted and the IC_50_ was calculated, giving a value of 12.18 µg/mL (Figure 10C).

#### 3.3.2. Cytotoxic Activity of DOX and CNS Drugs in MCF-7 Cancer Cells

We next evaluated if CNS drugs would also demonstrate anti-cancer potential against breast cancer cells and compared them with DOX, a very potent chemotherapeutic drug commonly used for breast cancer therapy. Contrary to HT-29 cells, CNS drugs did not demonstrate better anti-tumour potency than the antineoplastic drugs against MCF-7 cells (Figure 11A). This was also seen by analysing the IC_50_ values obtained for these drugs in this cell line. DOX displayed a greater anti-cancer effect than all CNS drugs in breast cancer cells, with an IC50 of 0.17 µM. For fluphenazine, sertraline and thioridazine the IC_50_ values obtained were 2.68, 5.72 and 2.22, respectively. Among all CNS drugs tested, fluoxetine displayed the highest IC50 value of 7.78 µM which correlates with the lower potency of the drug (Figure 11B).

#### 3.3.3. In Vitro Anti-Cancer Activity of Honeybee Venom Combined with DOX and CNS Drugs

Following the previously described methodologies for HT-29, we next treated MCF-7 breast cancer cells with combinations of DOX and CNS drugs plus honeybee venom. The combination model included the treatment of cells with increasing concentrations of honeybee venom plus the IC_50_ value of each drug. The combination of DOX with bee venom did not result in significant improvements compared to DOX and bee venom alone, as demonstrated in the morphological analysis (Figure 12A), MTT (Figure 12B) and SRB (Figure 12C) assays.

Co-treatment of fluphenazine and 6.25 µg/mL of honeybee venom resulted in enhanced cell viability reduction, compared to both drugs alone. Higher concentrations of honeybee venom and fluphenazine only resulted in statistical differences compared to fluphenazine alone, demonstrating that the anti-cancer activity of the combination was mainly due to the effect of honeybee venom (Figure 13B). These results are supported by the morphological analysis (Figure 13A). The SRB assay did not demonstrate significant differences in protein cellular content (Figure 13C).

The combination of fluoxetine and sertraline with increasing concentrations of honeybee venom resulted in similar results (Figure 14 and Figure 15, respectively). It was found that these combinations were statistically different from single treatments and effectively reduced MCF-7 cell viability and cellular protein content when using 6.25 µg/mL honeybee venom in the combination with the repurposed drug. These results were also supported by microscopy analysis. When using higher concentrations of bee venom, the observed result was only statistically different from the repurposed drug alone, having an anti-cancer effect similar to bee venom alone.

A low number of viable cells and cellular aggregates was seen in the combination treatment of thioridazine and bee venom (Figure 16A). MTT results demonstrate that co-treatment of thioridazine and 6.25 µg/mL honeybee venom significantly reduced breast cancer cell viability in relation to both drugs alone (Figure 16B). Protein content was significantly lower than bee venom only for the lowest concentration, being similar to bee venom in concentrations above 25 µg/mL (Figure 16C).

#### 3.3.4. Evaluation of Drug Interactions in MCF-7 Cells Treated with Honeybee Venom Combined with DOX and CNS Drugs

The potential for synergistic cytotoxic activity between DOX, CNS drugs and honeybee venom was first evaluated using the Chou–Talalay reference (Figure 17). The produced effect (Fa) and CI values for each combination are represented in Table 2. Based on the CI values, all drug combinations with CNS drugs indicated synergism for concentrations below 25 µg/mL of honeybee venom. The evaluation of drug interactions in cells treated with the combination of honeybee venom with DOX indicated mostly antagonism, except for DOX + 25 µg/mL honeybee venom.

The evaluation of drug interactions using the SynergyFinder software by other four reference models (ZIP, HSA; Bliss and Loewe) also demonstrated that the combination of DOX plus honeybee venom was predominantly antagonistic (Appendix A), while combinations with CNS drugs were mostly additive (Appendix A). For the combined pair of thioridazine with honeybee venom, the ZIP synergy plot was not calculated by the software.

## 4. Discussion

Colorectal and breast cancers are among the most prevalent types of cancer worldwide. Their high prevalence and mortality rates are due to the low efficacy of available treatments, where chemotherapy plays an important role. The development of new drugs for cancer therapy is not viable, as this is a process that takes a lot of time, money and usually results in low success rates [5]. Most chemotherapeutic drugs available in the market have problems related to toxicity and severe side effects [27]. Novel ideas and pharmaceutical strategies to overcome the problems related to chemotherapy and associated with the development of new drugs are therefore needed. In this work, our group took advantage of two strategies to develop a new pharmacological system for colorectal and breast cancer therapy: drug combination and drug repurposing. These strategies have already gained the attention of many research groups in recent years and results suggest they can be very promising.

Drug repurposing uses drugs that are already in the market (i.e., approved by the FDA) for new therapeutic indications, being advantageous over the development of new drugs since repurposed drugs have known safety and toxicity profiles. This saves time and money while increasing the probability of these drugs entering clinical trials [28]. Drug combination involves the administration of a mix of two or more drugs [29], allowing to overcome intratumoural and intertumoural heterogeneity, the main obstacles in cancer therapy. In recent years, several studies have demonstrated that they are indeed more effective than monotherapy [30,31,32,33,34,35].

Our research group has some experience in the fields of drug combination and drug repurposing [14,36,37], so we have designed a new combination model to evaluate if the combination of a natural compound, such as honeybee venom, could help improve the anti-cancer activity of 5-FU and DOX, two common antineoplastic drugs already used in the therapy of colon and breast cancer, respectively. We also evaluated if CNS drugs, such as fluphenazine, fluoxetine, sertraline and thioridazine, could be effectively repurposed for these two types of cancer, and if honeybee venom could also help improve their anti-cancer activity in two cell lines: MCF-7 breast and HT-29 colon cells. Recently, our group has already reported these drugs being effective against cancer cells [38].

Honeybee venom composition is very complex with at least 18 active compounds, including peptides, enzymes and amines [19]. Melittin was the main compound found in the *Apis mellifera iberiensis* venom, followed by phospholipase A2 and apamin. The composition can vary substantially according to factors concerning the bee such as age, strain and caste, and other factors such as season and collection methods [39]. The main compound, melittin, is a 26 amino acid peptide with 2840 Da, with water-soluble, linear, cationic, hemolytic and amphipathic properties [20]. Melittin has been recognized as an antiviral, antibacterial, antifungal, anti-parasitic and anti-tumour peptide due to its action as a non-selective cytolytic peptide that physically and chemically disrupts all prokaryotic and eukaryotic cell membranes [40].

We successfully chemically characterized honeybee venom from *Apis mellifera iberiensis* and found that the peptide melittin was the most abundant compound with a concentration of 65.6 ± 0.3%, followed by the enzyme phospholipase A2 with 4.0 ± 0.2% and the peptide apamin with 1.2 ± 0.1%. We next evaluated the cytotoxic effect of bee venom, DOX, 5-FU and four different CNS drugs (fluphenazine, fluoxetine, sertraline and thioridazine) in HT-29 colon and MCF-7 breast cancer cells. 5-FU and DOX were used in HT-29 and MCF-7 cells, respectively, as a comparison to the anti-cancer potential of the repurposed drugs. First, these cells were incubated with increasing concentrations of each compound and their metabolic activity was measured by MTT and SRB assays, in order to infer cell viability and cellular protein content. The cytotoxic evaluation of honeybee venom revealed that this compound has anti-cancer activity in both cell lines, showing great potential in reducing MCF-7 cell viability to a higher extent than HT-29 cells for lower concentrations (under 25 µg/mL). For concentrations above 50 µg/mL, honeybee venom was more effective in reducing the number of HT-29 viable cells (Appendix A). The IC_50_ of honeybee venom obtained for HT-29 and MCF-7 was 24.25 and 12.18 µg/mL, respectively. Together, these results support the anti-cancer potential of honeybee venom.

Repurposed drugs, fluphenazine, fluoxetine, sertraline and thioridazine, demonstrated a higher ability in reducing the viability of HT-29 cells than 5-FU. The IC_50_ obtained was 3.56, 1.86, 6.12, 2.45 and 4.26 µM for 5-FU, fluphenazine, fluoxetine, sertraline and thioridazine, respectively. The opposite was found when comparing these drugs with the cytotoxic activity of DOX in MCF-7 cells, where it was found that DOX had a higher potency than CNS drugs. The IC_50_ obtained were 0.17, 2.68, 7.78, 2.22 and 5.72 µM for DOX, fluphenazine, fluoxetine, sertraline and thioridazine, respectively. These results demonstrate that CNS drugs have a higher anti-cancer potential in colon cancer cells than in breast cancer cells.

Next, we combined variable doses (6.25–100 µg/mL) of honeybee venom with a fixed dose (IC_50_ value) of chemotherapeutic and CNS drugs in both cell lines. Comparing the cytotoxic activity of these combinations, it was found that bee venom combined with 5-FU and fluphenazine in HT-29 cells resulted in less cytotoxic effects compared to the co-treatment of fluoxetine, sertraline and thioridazine plus bee venom, which resulted in less than 15% of viable cells for the whole range of concentrations (Appendix A). For lower concentrations, combined treatments resulted in better anti-cancer activities than both drugs alone. It was also found that the combination of repurposed drugs with honeybee venom resulted in synergistic interactions in some drug pairs with a CI under 1. Results regarding combinations in MCF-7 cells demonstrated an increased ability to reduce the number of viable cells than with HT-29 cells. Nevertheless, combinations with repurposed drugs resulted in better anti-cancer efficacies than with DOX, notably for lower concentrations, with indications of synergism and a CI < 1 (Appendix A). Combinations of fluoxetine and thioridazine plus honeybee venom resulted in less than 40% of viable cells for all ranges of concentrations. These results support the idea that the combination of honeybee venom with repurposed drugs and chemotherapeutic agents can help improve their anti-cancer activities, especially for lower concentrations, in both cell lines. These combinations also demonstrate greater potential for applications in colon cancer cells than in breast cancer therapy, with an indication of more synergistic interactions in HT-29 cancer cells. Drug interaction evaluations by SynergyFinder also demonstrated that the choice of the reference model can give slightly different results regarding the synergy evaluation of drug combinations, although these reference models produce similar results most of the time.

Together, these results demonstrate that chemotherapeutic and repurposed drugs combined with honeybee venom, with peptide MEL as principal compound, are promising compounds for the chemotherapy of colorectal and breast cancer, being good candidates for employment in combination therapies.

Further analysis of synergistic combination data indicated that it can result from a very effective interaction with the main constituent of the honeybee venom, the peptide MEL. To test some of the sequence-based functionalities, the calculation of several peptide physicochemical properties was run to observe its components, such as the identification of a dominant property, or by contrast, realising how promiscuous were the drugs’ proprieties to bind peptides with different characteristics. The average values from Appendix A are here included to describe general trends of the observables for peptide alone. However, we believe that the combination of chemotherapeutic and CNS drugs with peptide MEL has a wide range of value properties, including hydrophobic to hydrophilic sequences, which is characteristic of the receptor’s promiscuity.

Following these promising in vitro results, more research should be performed, including mechanistic studies, to evaluate the anti-cancer mechanisms underlying these combinations and their possible application in animal models and clinical trials. Overall, the present study supports the enormous potential of honeybee venom for breast and colorectal cancer treatments both alone and in combination with chemotherapy and CNS repurposed drugs.

## Figures and Tables

**Figure 1 pharmaceutics-14-00511-f001:**
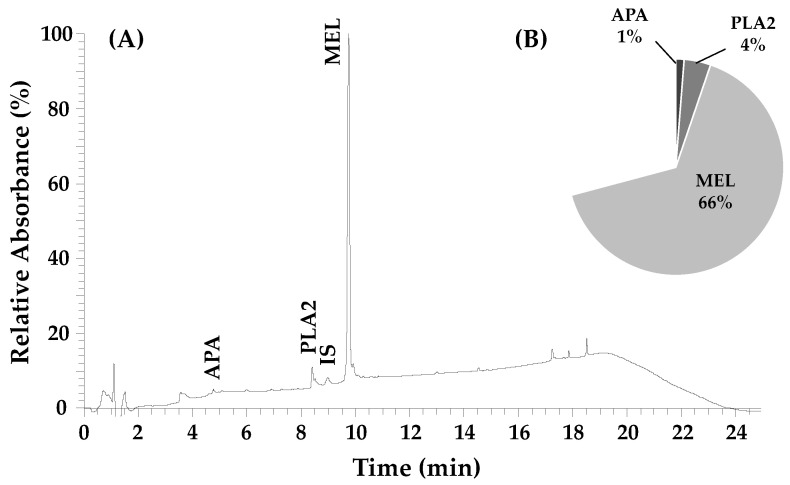
(**A**) Chromatographic profile of Apis mellifera iberiensis venom at 220 nm; (**B**) quantification of the main compounds of the honeybee venom; APA—apamim; PLA2—phospholipase A2; IS—internal standard (cytochrome c, 25 µg/mL); MEL—melittin.

**Figure 2 pharmaceutics-14-00511-f002:**
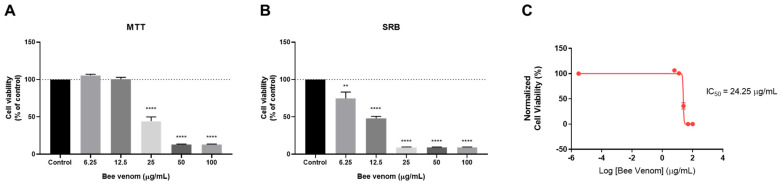
Cell viability of HT-29 colon cancer cells treated with honeybee venom. (**A**) MTT assay (**B**) SRB assay and (**C**) the dose–response curve of HT-29 cells cultured in 96-well plates and treated with increasing concentrations (6.25–100 µg/mL) of honeybee venom for 48 h. The dose–response curve was obtained by non-linear regression after normalization of cell viability values. Each point represents the mean ± SEM relative to the control cells (0.1% DMSO) of three independent experiences. ** statistically significant vs. control at *p* < 0.01; **** statistically significant vs. control at *p* < 0.0001.

**Figure 3 pharmaceutics-14-00511-f003:**
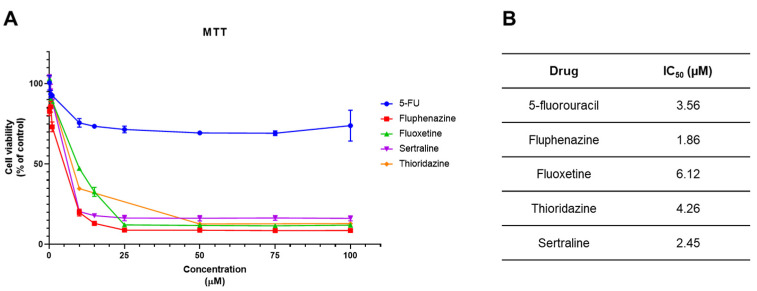
Cytotoxic effect of 5-FU and CNS drugs in HT-29 colon cancer cells. (**A**) MTT results of HT-29 cells treated with increasing concentrations of 5-FU and four different CNS drugs. HT-29 cells were cultured in 96-well plates and treated with increasing concentrations (0.1–100 µM) of each drug for 48 h. (**B**) IC_50_ values obtained for 5-FU and CNS drugs in HT-29 cells. These values were obtained by plotting dose–response curves for each drug using non-linear regression after normalization of cell viability values. Each point represents the mean ± SEM relative to the control cells (0.1% DMSO) of three independent experiences.

**Figure 4 pharmaceutics-14-00511-f004:**
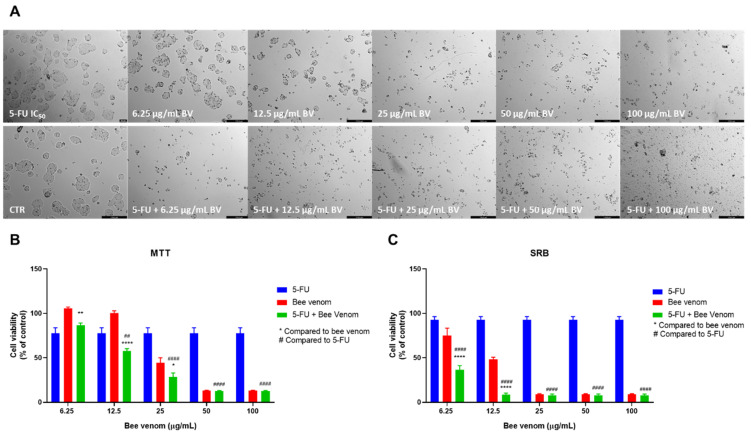
Effect of honeybee venom combined with 5-FU in HT-29 colon cancer cells. (**A**) Morphological analysis; (**B**) MTT and (**C**) SRB assays. HT-29 cells were cultured in 96-well plates and treated with each combination for 48 h. Cell viabilities were determined after the final treatment. Each point represents the mean ± SEM relative to the control cells (0.1% DMSO) of three independent experiences. * Statistically significant vs. bee venom at *p* < 0.05; ** Statistically significant vs. bee venom at *p* < 0.01; **** Statistically significant vs. bee venom at *p* < 0.0001; ^##^ Statistically significant vs. 5-FU at *p* < 0.01; ^####^ Statistically significant vs. 5-FU at *p* < 0.0001. Scale bar: 179.3 µm; BV: bee venom.

**Figure 5 pharmaceutics-14-00511-f005:**
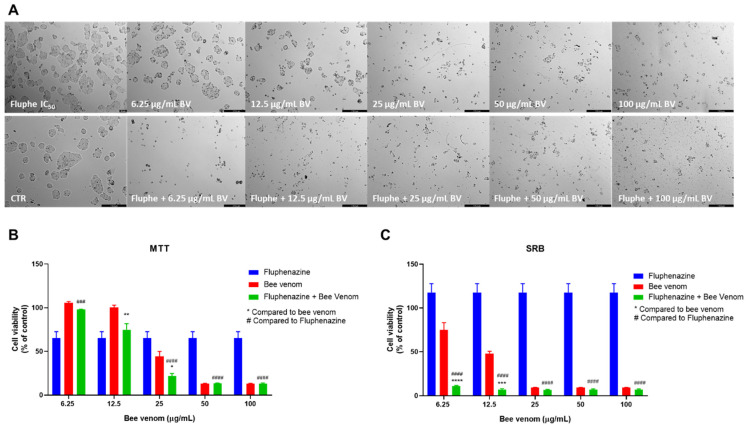
Effect of honeybee venom combined with fluphenazine in HT-29 colon cancer cells. (**A**) Morphological analysis; (**B**) MTT and (**C**) SRB assays. HT-29 cells were cultured in 96-well plates and treated with each combination for 48 h. Cell viabilities were determined after the final treatment. Each point represents the mean ± SEM relative to the control cells (0.1% DMSO) of three independent experiences. * Statistically significant vs. bee venom at *p* < 0.05; ** Statistically significant vs. bee venom at *p* < 0.01; *** Statistically significant vs. bee venom at *p* < 0.001; **** Statistically significant vs. bee venom at *p* < 0.0001; ^###^ Statistically significant vs. fluphenazine at *p* < 0.001; ^####^ Statistically significant vs. fluphenazine at *p* < 0.0001. Scale bar: 179.3 µm; BV: bee venom.

**Figure 6 pharmaceutics-14-00511-f006:**
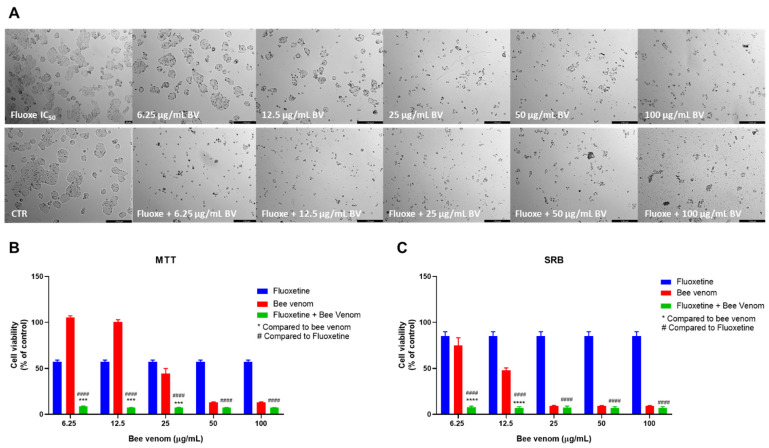
Effect of honeybee venom combined with fluoxetine in HT-29 colon cancer cells. (**A**) Morphological analysis; (**B**) MTT and (**C**) SRB assays. HT-29 cells were cultured in 96-well plates and treated with each combination for 48 h. Cell viabilities were determined after the final treatment. Each point represents the mean ± SEM relative to the control cells (0.1% DMSO) of three independent experiences. *** Statistically significant vs. bee venom at *p* < 0.001; **** Statistically significant vs. bee venom at *p* < 0.0001; ^####^ Statistically significant vs. fluoxetine at *p* < 0.0001. Scale bar: 179.3 µm; BV: bee venom.

**Figure 7 pharmaceutics-14-00511-f007:**
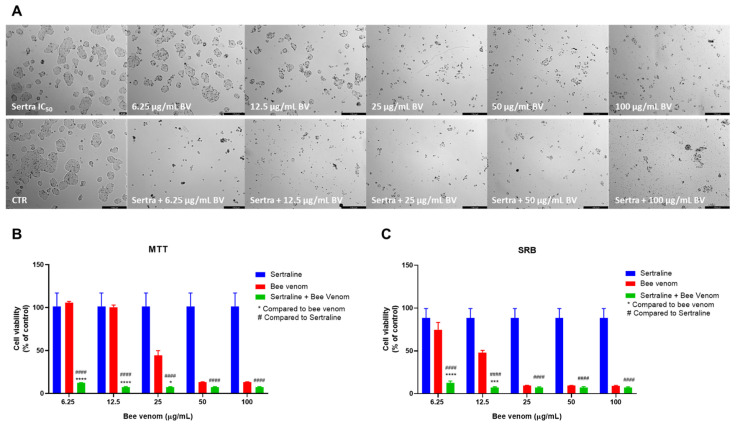
Effect of honeybee venom combined with sertraline in HT-29 colon cancer cells. (**A**) Morphological analysis; (**B**) MTT and (**C**) SRB assays. HT-29 cells were cultured in 96-well plates and treated with each combination for 48 h. Cell viabilities were determined after the final treatment. Each point represents the mean ± SEM relative to the control cells (0.1% DMSO) of three independent experiences. * Statistically significant vs. bee venom at *p* < 0.05; *** Statistically significant vs. bee venom at *p* < 0.001; **** Statistically significant vs. bee venom at *p* < 0.0001; ^####^ Statistically significant vs. sertraline at *p* < 0.0001. Scale bar: 179.3 µm; BV: bee venom.

**Figure 8 pharmaceutics-14-00511-f008:**
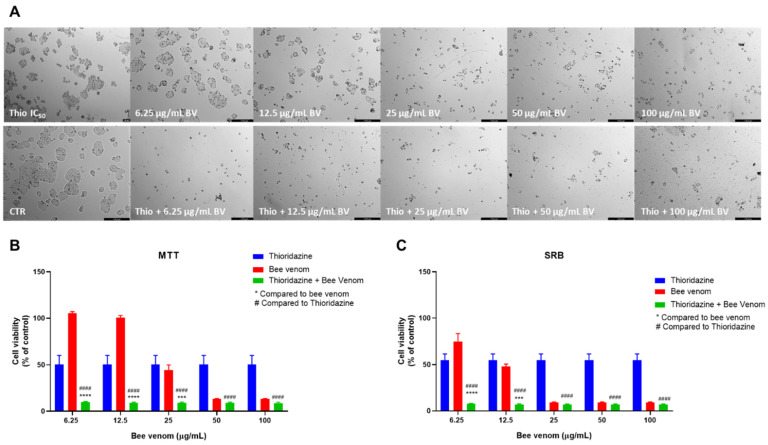
Effect of honeybee venom combined with thioridazine in HT-29 colon cancer cells. (**A**) Morphological analysis; (**B**) MTT and (**C**) SRB assays. HT-29 cells were cultured in 96-well plates and treated with each combination for 48 h. Cell viabilities were determined after the final treatment. Each point represents the mean ± SEM relative to the control cells (0.1% DMSO) of three independent experiences. *** Statistically significant vs. bee venom at *p* < 0.001; **** Statistically significant vs. bee venom at *p* < 0.0001; ^####^ Statistically significant vs. thioridazine at *p* < 0.0001. Scale bar: 179.3 µm; BV: bee venom.

**Figure 9 pharmaceutics-14-00511-f009:**
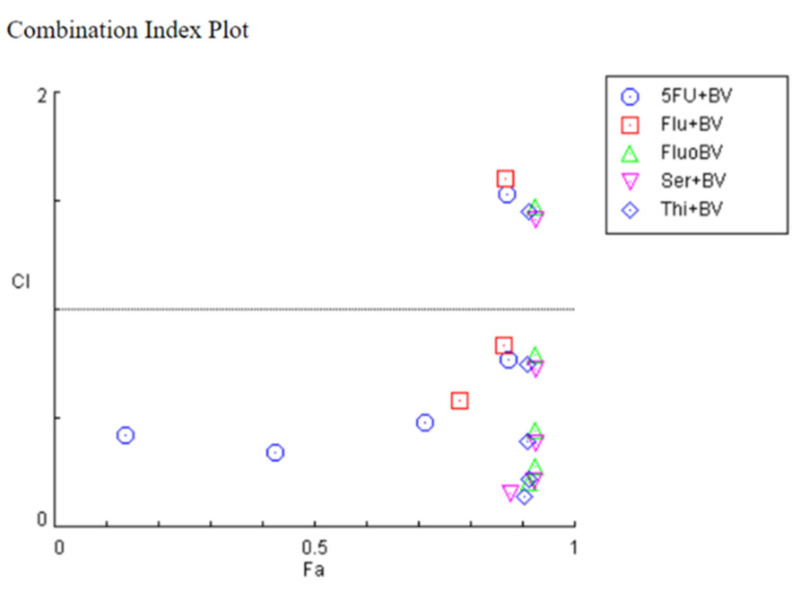
Fa-CI plot of combined treatments of 5-FU and CNS drugs combined with honeybee venom on HT-29 colon cancer cells. The combination index was calculated using CompuSyn software. CI < 1, =1, and >1 indicates synergistic, additive, and antagonistic effects, respectively. BV: bee venom; 5FU: 5-fluorouracil; Flu: fluphenazine; Fluo: fluoxetine; Ser: sertraline; Thi: thioridazine.

**Figure 10 pharmaceutics-14-00511-f010:**
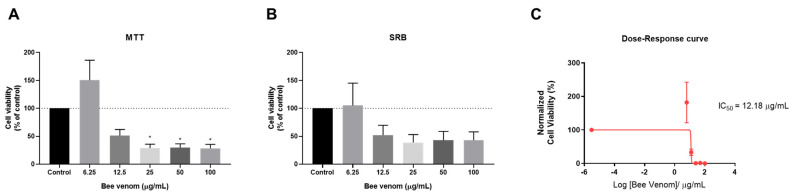
Cell viability of MCF-7 breast cancer cells treated with honeybee venom. (**A**) MTT assay, (**B**) SRB assay and (**C**) the dose–response curve of MCF-7 cells cultured in 96-well plates and treated with increasing concentrations (6.25–100 µg/mL) of honeybee venom for 48 h. The dose–response curve was obtained by non-linear regression after normalization of cell viability values. Each point represents the mean ± SEM relative to the control cells (0.1% DMSO) of three independent experiences. * Statistically significant vs. control at *p* < 0.05.

**Figure 11 pharmaceutics-14-00511-f011:**
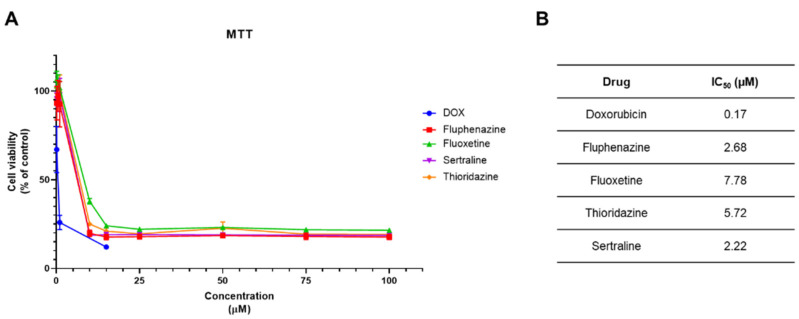
Cytotoxic effect of DOX and CNS drugs in MCF-7 breast cancer cells. (**A**) MTT results of MCF-7 cells treated with increasing concentrations of DOX and four different CNS drugs. MCF-7 cells were cultured in 96-well plates and treated with increasing concentrations (0.1–100 µM) of each drug for 48 h, except for DOX which was tested from 0.1–15 µM. (**B**) IC_50_ values obtained for DOX and CNS drugs in MCF-7 cells. These values were obtained by plotting dose–response curves for each drug using non-linear regression after normalization of cell viability values. Each point represents the mean ± SEM relative to the control cells (0.1% DMSO) of three independent experiences.

**Figure 12 pharmaceutics-14-00511-f012:**
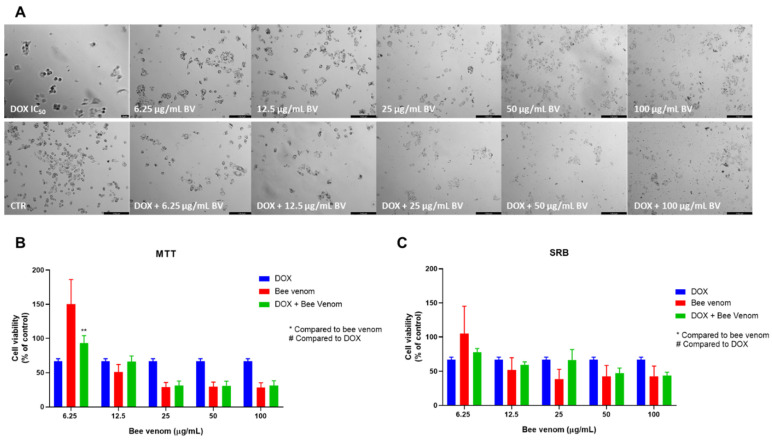
Effect of honeybee venom combined with DOX in MCF-7 breast cancer cells. (**A**) Morphological analysis; (**B**) MTT and (**C**) SRB assays. MCF-7 cells were cultured in 96-well plates and treated with each combination for 48 h. Cell viabilities were determined after the final treatment. Each point represents the mean ± SEM relative to the control cells (0.1% DMSO) of three independent experiences. ** Statistically significant vs. bee venom at *p* < 0.01; Scale bar: 179.3 µm; BV: bee venom.

**Figure 13 pharmaceutics-14-00511-f013:**
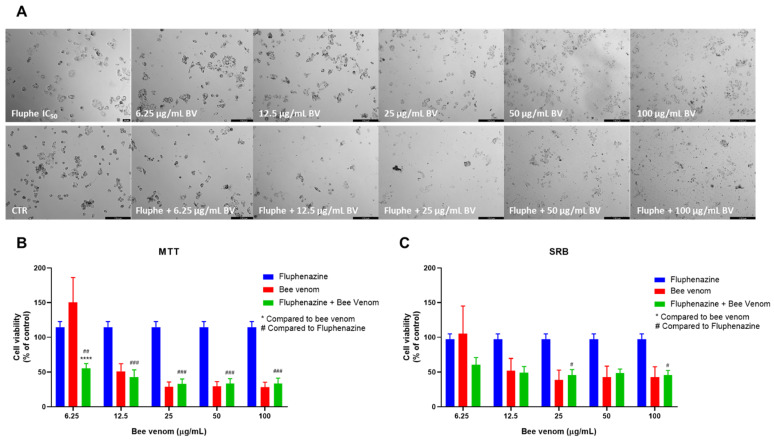
Effect of honeybee venom combined with fluphenazine in MCF-7 breast cancer cells. (**A**) Morphological analysis; (**B**) MTT and (**C**) SRB assays. MCF-7 cells were cultured in 96-well plates and treated with each combination for 48 h. Cell viabilities were determined after the final treatment. Each point represents the mean ± SEM relative to the control cells (0.1% DMSO) of three independent experiences. **** Statistically significant vs. bee venom at *p* < 0.0001; ^#^ Statistically significant vs. fluphenazine at *p* < 0.05; ^##^ Statistically significant vs. fluphenazine at *p* < 0.01; ^###^ Statistically significant vs. fluphenazine at *p* < 0.001. Scale bar: 179.3 µm; BV: bee venom.

**Figure 14 pharmaceutics-14-00511-f014:**
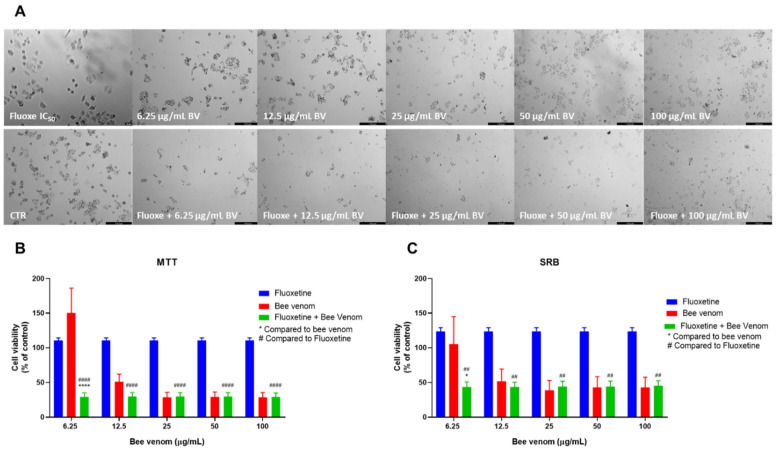
Effect of honeybee venom combined with fluoxetine in MCF-7 breast cancer cells. (**A**) Morphological analysis; (**B**) MTT and (**C**) SRB assays. MCF-7 cells were cultured in 96-well plates and treated with each combination for 48 h. Cell viabilities were determined after the final treatment. Each point represents the mean ± SEM relative to the control cells (0.1% DMSO) of three independent experiences. * Statistically significant vs. bee venom at *p* < 0.05; **** Statistically significant vs. bee venom at *p* < 0.0001; ^##^ Statistically significant vs. fluoxetine at *p* < 0.01; ^####^ Statistically significant vs. fluoxetine at *p* < 0.0001. Scale bar: 179.3 µm; BV: bee venom.

**Figure 15 pharmaceutics-14-00511-f015:**
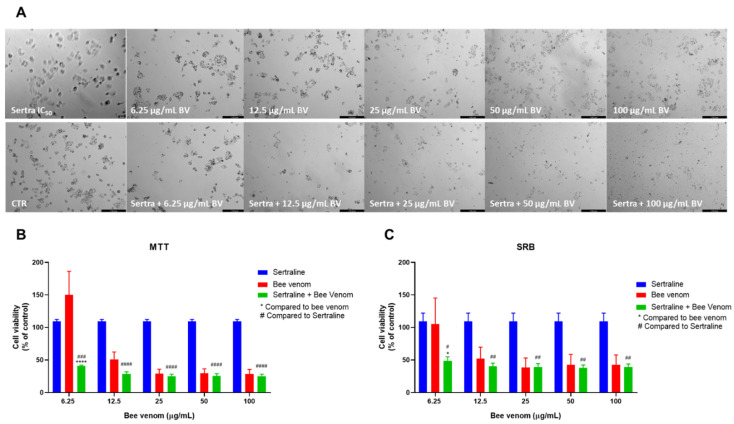
Effect of honeybee venom combined with sertraline in MCF-7 breast cancer cells. (**A**) Morphological analysis; (**B**) MTT and (**C**) SRB assays. MCF-7 cells were cultured in 96-well plates and treated with each combination for 48 h. Cell viabilities were determined after the final treatment. Each point represents the mean ± SEM relative to the control cells (0.1% DMSO) of three independent experiences. * Statistically significant vs. bee venom at *p* < 0.05; **** Statistically significant vs. bee venom at *p* < 0.0001; ^#^ Statistically significant vs. sertraline at *p* < 0.05; ^##^ Statistically significant vs. sertraline at *p* < 0.01; ^###^ Statistically significant vs. sertraline at *p* < 0.001; ^####^ Statistically significant vs. sertraline at *p* < 0.0001. Scale bar: 179.3 µm; BV: bee venom.

**Figure 16 pharmaceutics-14-00511-f016:**
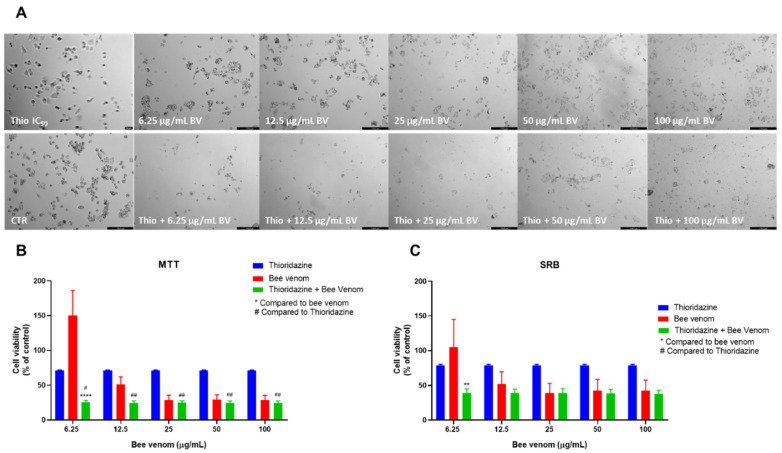
Effect of honeybee venom combined with thioridazine in MCF-7 breast cancer cells. (**A**) Morphological analysis; (**B**) MTT and (**C**) SRB assays. MCF-7 cells were cultured in 96-well plates and treated with each combination for 48 h. Cell viabilities were determined after the final treatment. Each point represents the mean ± SEM relative to the control cells (0.1% DMSO) of three independent experiences. ** Statistically significant vs. bee venom at *p* < 0.01; **** Statistically significant vs. bee venom at *p* < 0.0001; ^#^ Statistically significant vs. thioridazine at *p* < 0.05; ^##^ Statistically significant vs. thioridazine at *p* < 0.01. Scale bar: 179.3 µm; BV: bee venom.

**Figure 17 pharmaceutics-14-00511-f017:**
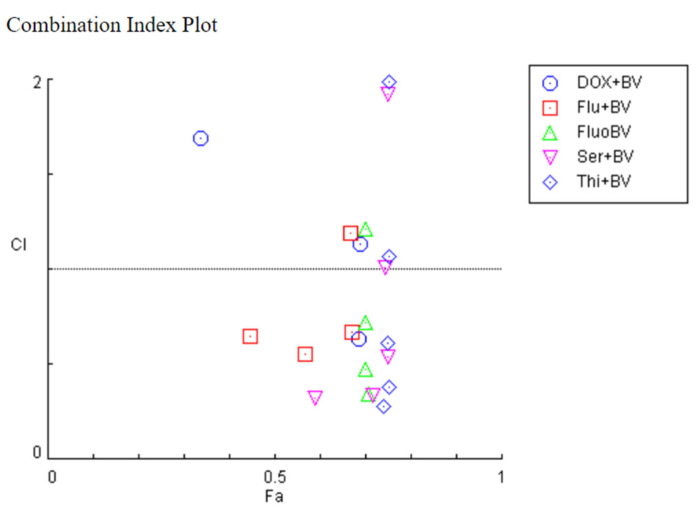
Fa-CI plot of combined treatments of DOX and CNS drugs combined with honeybee venom on MCF-7 breast cancer cells. The combination index was calculated using CompuSyn software. CI < 1, =1, and >1 indicates synergistic, additive, and antagonistic effects, respectively. BV: bee venom; DOX: doxorubicin; Flu: fluphenazine; Fluo: fluoxetine; Ser: sertraline; Thi: thioridazine.

**Table 1 pharmaceutics-14-00511-t001:** The nature of drug interactions in HT-29 colon cancer cells treated with CNS drugs and 5-FU combined with honeybee venom.

Drug A	Dose A (µM)	Sample B	Dose B (µg/mL)	Effect (Fa)	CI value	Interaction
5-FU	3.56	Honeybee Venom	6.25	0.1364	0.42	Synergism
			12.5	0.4246	0.34	Synergism
			25	0.7125	0.47	Synergism
			50	0.8731	0.77	Synergism
			100	0.8723	1.53	Antagonism
Fluphenazine	1.86	Honeybee Venom	6.25	0.0193	>100	Antagonism
			12.5	0.2529	4.39	Antagonism
			25	0.7797	0.58	Synergism
			50	0.866	0.83	Synergism
			100	0.867	1.60	Antagonism
Fluoxetine	6.12	Honeybee Venom	6.25	0.9125	0.20	Synergism
			12.5	0.9255	0.27	Synergism
			25	0.9244	0.44	Synergism
			50	0.9252	0.79	Synergism
			100	0.9256	1.47	Antagonism
Sertraline	2.45	Honeybee Venom	6.25	0.877	0.15	Synergism
			12.5	0.924	0.21	Synergism
			25	0.9249	0.38	Synergism
			50	0.9246	0.72	Synergism
			100	0.9247	1.41	Antagonism
Thioridazine	4.26	Honeybee Venom	6.25	0.9034	0.13	Synergism
			12.5	0.9122	0.21	Synergism
			25	0.911	0.39	Synergism
			50	0.912	0.75	Synergism
			100	0.913	1.45	Antagonism

**Table 2 pharmaceutics-14-00511-t002:** The nature of drug interactions in MCF-7 breast cancer cells treated with CNS drugs and DOX combined with honeybee venom.

Drug A	Dose A (µM)	Sample B	Dose B (µg/mL)	Effect (Fa)	CI Value	Interaction
DOX	0.17	Honeybee Venom	6.25	0.0656	30.74	Antagonism
			12.5	0.3388	1.69	Antagonism
			25	0.6877	0.63	Synergism
			50	0.6897	1.13	Antagonism
			100	0.6851	2.15	Antagonism
Fluphenazine	2.68	Honeybee Venom	6.25	0.4464	0.64	Synergism
			12.5	0.5691	0.55	Synergism
			25	0.6707	0.67	Synergism
			50	0.667	1.19	Antagonism
			100	0.6646	2.23	Antagonism
Fluoxetine	7.78	Honeybee Venom	6.25	0.7064	0.34	Synergism
			12.5	0.7025	0.47	Synergism
			25	0.7016	0.72	Synergism
			50	0.7022	1.21	Antagonism
			100	0.7076	2.18	Antagonism
Sertraline	2.22	Honeybee Venom	6.25	0.5892	0.32	Synergism
			12.5	0.716	0.33	Synergism
			25	0.7492	0.53	Synergism
			50	0.7442	1.01	Antagonism
			100	0.7497	1.92	Antagonism
Thioridazine	5.72	Honeybee Venom	6.25	0.7398	0.27	Synergism
			12.5	0.7523	0.38	Synergism
			25	0.7505	0.61	Synergism
			50	0.7545	1.06	Antagonism
			100	0.7545	1.98	Antagonism

## Data Availability

Not applicable.

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
