# Peer review of "Honeybee Venom Synergistically Enhances the Cytotoxic Effect of CNS Drugs in HT-29 Colon and MCF-7 Breast Cancer Cell Lines"

_pharmaceutics, 2022, doi:10.3390/pharmaceutics14030511_

Round 1

Reviewer 1 Report

The authors have satisfactorily responded to all the questions and made the necessary changes to the manuscript. I have no further questions and suggest the acceptance of the revised manuscript.

Author Response

Dear Editor,

Thank you for giving us another opportunity to answer the major concerns suggested by the reviewers. We greatly appreciate the time and effort that you and the reviewers have again dedicated to giving feedback to our manuscript. The answers to the reviewers are next provided. Changes in the main text and images were highlighted in red.

# Reviewer 1

The authors have satisfactorily responded to all the questions and made the necessary changes to the manuscript. I have no further questions and suggest the acceptance of the revised manuscript.

We thank the reviewer for the time spent reviewing our manuscript and for the positive feedback.

Reviewer 2 Report

The manuscript by Duarte et al. presents the synergistic effect of CNSs drugs on the cytotoxicity of honey bee venom against HT-29 and MCF-7 cancer cells. The manuscript was not well organized and prepared with a lot of track changes showing up. I am thinking there might be an error when the manuscript was converted from a word document to a pdf file or the author submitted a wrong manuscript version. However, the research outcomes were very good and beneficial for further anticancer drug studies. I recommend that the manuscript should be accepted for publication in the Pharmaceutics journal after addressing following comments.

  1. Most of the figures in the manuscript excepting figures 1 and 28 have low-resolutions which cause difficulties to read the data. Therefore, the authors must improve Figure quality.
  2. Page 10, Figure 3: I am concerned about the dose-response curve of 5-FU. The curve indicated HT-29 cells were resistant to 5-FU as 75% of HT-29 cells remained alive when the concentration of 5-FU increased from 25–100 µM. The authors need to repeat this experiment to confirm the cytotoxicity of 5-FU and its IC50
  3. Page 28, Figure 15C: One concentration (12 µg/mL?) of honey bee venom had 200% cell viability which was unacceptable without any explanation. The percentage of cell viability at this concentration is important to determine a correct IC50 value of bee venom. I would suggest the authors repeat this experiment.
  4. There are too many figures in the manuscript which will confuse readers. I would suggest the authors select the key figures to show in the manuscript and move others in a supporting information.

The authors should mention a reason why using both MTT and SRB assays to evaluate the cell viability in the main text.

Author Response

Dear Editor,

Thank you for giving us another opportunity to answer the major concerns suggested by the reviewers. We greatly appreciate the time and effort that you and the reviewers have again dedicated to giving feedback to our manuscript. The answers to the reviewers are next provided. Changes in the main text and images were highlighted in red.

# Reviewer 2

The manuscript by Duarte et al. presents the synergistic effect of CNSs drugs on the cytotoxicity of honeybee venom against HT-29 and MCF-7 cancer cells. The manuscript was not well organized and prepared with a lot of track changes showing up. I am thinking there might be an error when the manuscript was converted from a word document to a pdf file or the author submitted a wrong manuscript version. However, the research outcomes were very good and beneficial for further anticancer drug studies. I recommend that the manuscript should be accepted for publication in the Pharmaceutics journal after addressing following comments.

We thank the reviewer for the time spent reviewing our manuscript and for the positive feedback. Indeed, there might have been an error converting the manuscript to the pdf file as the word file is correct and with no track changes shown. We thank the reviewer for the opportunity to resubmit another revised version.

  1. Most of the figures in the manuscript excepting figures 1 and 28 have low-resolutions which cause difficulties to read the data. Therefore, the authors must improve Figure quality.

We thank the reviewer for the observation. Indeed, the conversion to the PDF file caused a loss in image resolution. Nevertheless, we have changed all figures to improve their resolution.

  1. Page 10, Figure 3: I am concerned about the dose-response curve of 5-FU. The curve indicated HT-29 cells were resistant to 5-FU as 75% of HT-29 cells remained alive when the concentration of 5-FU increased from 25–100 µM. The authors need to repeat this experiment to confirm the cytotoxicity of 5-FU and its IC50.

We understand the reviewer concern about the obtained IC50 value. We have already repeated these experiments for other studies in progress and have obtained similar results in HT-29 colon cancer cells:

We found in four independent experiments that 5-FU reaches a plateau of inhibition and increasing the concentration will not improve its cytotoxic effect. All IC50 values used throughout the manuscript were obtained by curve plots using the GraphPad software. Also, we would like to mention that differences between results from MTT assays and the obtained IC50 value can be explained by the difference between relative and absolute IC50 values. The values represented in our manuscript are relative IC50, which is the most common IC50 definition. This value represents the concentration that reduces the cell viability by 50% between the top and the bottom plateaus of the obtained dose-response curve, after the normalization of the data and nonlinear regression analysis. On the other hand, the absolute value of IC50 represents the value between 0% of cell death (blank) and 100% of cell death (control value (NS) that kills 100% of the cells, usually a high concentration of the drug), which is rarely used in a pharmacological context. So, the data input in dose-response curves is the normalized values of cell viability and not the raw data obtained on MTT studies.

  1. Page 28, Figure 15C: One concentration (12 µg/mL?) of honey bee venom had 200% cell viability which was unacceptable without any explanation. The percentage of cell viability at this concentration is important to determine a correct IC50 value of bee venom. I would suggest the authors repeat this experiment.

We understand the point of view of the reviewer. This dose-response curve was obtained based on the MTT results and the outlier point is correspondent to the concentration of 6.25 µg/mL and approximately 150% of cell viability in the MTT graph. The explanation of the 200% value seen in the dose-response graph is that this is a normalized value obtained from the normalization of the data represented in the MTT graph. The values represented in our manuscript are relative IC50 and this value represents the concentration that reduces the cell viability by 50% between the top and the bottom plateaus of the obtained dose-response curve, after the normalization of the data and nonlinear regression analysis. So, the data input in dose-response curves is the normalized values of cell viability and not the raw data obtained on MTT studies. Also, for lower concentrations of drugs, we have previously found that sometimes cells are not immediately killed by the treatment and try to adapt to the environment by activating survival mechanisms and increasing proliferating, which translates to higher viability values. In addition, the 150% of cell viability represented in the MTT results can be explained due to natural variations of cellular metabolism or because slightly more cells were seeded in that well due to small pipetting errors, so is it common to have samples that exceed the 100% of cell viability.

  1. There are too many figures in the manuscript which will confuse readers. I would suggest the authors select the key figures to show in the manuscript and move others in a supporting information.

We thank the reviewer for the suggestion and agree that there are many images in the manuscript. We have moved 13 figures regarding SynergyFinder simulations and all figures from the discussion section to the supporting material.

The authors should mention a reason why using both MTT and SRB assays to evaluate the cell viability in the main text.

We thank the reviewer for the suggestion. We have included this explanation in the main text: “These cell-based assays are among the most commonly used in pharmacological studies involving cancer cell lines to evaluate the anticancer effect of different chemotherapeutic agents and provide important information regarding the mitochondrial metabolism and cellular protein content. MTT assay measures the metabolic activity of cells, while SRB assay uses an acidic protein-dye (SRB) that binds electrostatically and pH-dependent to the basic amino acid residues of the proteins presented in the fixed cells, giving information about the cellular protein content.  These assays were also included in our previous combination study [14] and we have kept the same methodological design.”

Round 2

Reviewer 2 Report

The authors have addressed all my comments with reasonable explanations to improve the manuscript quality. I would suggest the revised manuscript to be accepted for publication.